# Detecting the PEX Like Domain of Matrix Metalloproteinase-14 (MMP-14) with Therapeutic Conjugated CNTs

**DOI:** 10.3390/bios12100884

**Published:** 2022-10-17

**Authors:** D. Vieira, J. Barralet, E. J. Harvey, G. Merle

**Affiliations:** 1Department of Experimental Surgery, Faculty of Medicine, McGill University, Montreal, QC H3G 2M1, Canada; daniela.vieira@mail.mcgill.ca (D.V.); jake.barralet@mcgill.ca (J.B.); edward.harvey@mcgill.ca (E.J.H.); 2Department of Surgery, McGill University, Montreal, QC H3G 1A4, Canada; 3Department of Chemical Engineering, Polytechnique Montreal, Montreal, QC H3T 1J4, Canada

**Keywords:** MMP-14, electrochemical detection, PEX domain, CNT, surface modification, cancer

## Abstract

Matrix metalloproteinases (MMPs) are essential proteins acting directly in the breakdown of the extra cellular matrix and so in cancer invasion and metastasis. Given its impact on tumor angiogenesis, monitoring MMP-14 provides strategic insights on cancer severity and treatment. In this work, we report a new approach to improve the electrochemical interaction of the MMP-14 with the electrode surface while preserving high specificity. This is based on the detection of the hemopexin (PEX) domain of MMP-14, which has a greater availability with a stable and low-cost commercial molecule, as a recognition element. This molecule, called NSC-405020, is specific of the PEX domain of MMP-14 within the binding pocket. Through the covalent grafting of the NSC-405020 molecule on carbon nanotubes (CNTs), we were able to detect and quantify MMP-14 using electrochemical impedance spectroscopy with a linear range of detection of 10 ng⋅mL^−1^ to 100 ng⋅mL^−1^, and LOD of 7.5 ng⋅mL^−1^. The specificity of the inhibitory small molecule was validated against the PEX domain of MMP-1. The inhibitor loaded CNTs system showed as a desirable candidate to become an alternative to the conventional recognition bioelements for the detection of MMP-14.

## 1. Introduction

Matrix metalloproteinases (MMPs) are zinc family protein involved in the breakdown of various components of the extracellular matrix in normal physiological processes, such as embryonic development, reproduction, and tissue remodeling, as well as in disease processes such as arthritis and cancer [1,2,3]. To date, there are 23 MMPs identified in humans [2,3]. According to their function and substrate specificity, MMPs are divided in the following groups: collagenases (MMPs-1, 8, 13), gelatinases (MMPs-2 and 9), stromelysins (MMPs-3, 10 and 11), matrilysins (MMPs-7 and 26), membrane-type (MMPs-14, 15, 16, 17, 23, 24 and 25) and others (MMPs-12, 19, 20, 22, 27 and 28) [4,5]. MMPs are the major proteases involved in the degradation of extracellular matrix [4,5,6]. MPPs present a typical structure consisting of at least three domains: predomain, propeptide, catalytic and hemopexin [4,5]. The expression of MMPs is maintained in the body at a constant low level; however, their abnormal expression has been associated with numerous diseases, including cancer [4,7,8]. Excessive expression of MMPs is reported in and around tumors and is associated with cancer stage, progression, metastasis, and mortality [9,10,11]. Given their important role in both physiological and pathological processes, MMPs have become valuable biomarkers to various specific cancers. Current methods to detect MMPs include liquid chromatography-mass spectroscopy [12,13,14], fluorescence resonance energy transfer [15,16,17], surface plasmon resonance [18,19] and enzyme-linked immunosorbent assay (ELISA) [20,21]. Despite high specificity and limit of detection, these techniques require qualified personal and high costs of maintenance and performance [3,22,23]. Compared to these molecular approaches, electrochemical techniques are simpler, faster, low-cost, and user-friendly, which makes them a more appropriate tool for the rapid detection of proteins [24]. The electrochemical detection is usually facilitated by a recognition element that will bind specifically to the target molecules [24,25,26]. Most typically detections are accomplished via biomolecules, such as antibody, enzymes, aptamers, nucleic acids, and peptides attached to gold/carbon substrates [24,26,27,28]; however, a crucial feature is not only the binding affinity but also the interfacial stability, where bioelements in terms of stability in terms of stability, are limited to physiological conditions and prone to irreversible denaturation and most of the case, expensive, affecting the activity of the sensors and limiting their use [25,29,30]. 

MMP-14, also known as MT1-MMP, is of particular interest in the field of oncology. It is an anchored membrane protein and has shown significant contribution in tumor angiogenesis by cleaving extra cellular matrix molecules as a matrix-degrading enzyme [4,10,11]. MMP-14 also coordinates key pro-angiogenic factors, such as VEGF, pro-TGF-β and endoglin, suggesting crucial role in vessel maturation and formation [8,10,31]. Because of the elevated MMP-14 expression observed in breast [32,33], head and neck [33], fibrosarcoma [33,34], prostate [34], gastric [35], bladder [36], ovarian [37] and brain [38] cancer, the detection and measurement of MMP-14 has an essential role in the diagnostic and the treatment direction of cancer The catalytic domain is usually the target domain for the electrochemical detection of MMPs because of communication between the redox active center (zinc) and the electrode surface. However, the proteolytic site is usually hidden within the insulating protein shell, making the electrochemical response very challenging to measure at the electrode [39]. An alternative to the catalytic approach is to use specific molecules that bind to the other domains, e.g., hemopexin (PEX). Except for MMPs-7, 23 and 26, all humans MMPs are expressed with a PEX-like domain [5,6]. PEX domain regulates key specific functions in different MMPs [40,41]; For example, PEX mediates the activation process of gelatinase (MMP-2) and collagenases (MMP-1 and 13); blocks the glycoproteins TIMPs (MMP-1, 2, 9 and 13) and clusterin (MMP-25), natural inhibitors of MMPs; assists on the homodimerization of MMP-1, 9 and 14; binds and cleavages different substrates, such as chemokines (MMP-2), C1q (MMP-14), IGFBP-3 (MMP-19), fibrinogen (MMP-2), etc.; and is crucial for the attachment of MMPs to the cell surface [41]. PEX domain demonstrates easy access and abundance in the protein structure when compared to the hidden zinc ion on the catalytic core of the protein [42]. 

Currently, only one approach has been employing inhibitory peptide to interact electrochemically with the PEX domain for the electrochemical detection of MMP-14 [22,43]. These peptide attached onto gold electrode exhibits a fairly good specificity towards MMP14 with a limit of detection (LOD) of 7ng·L^−1^ after 30 min, but some concerns associated with reduced shelf life (temperature sensitivity, denaturation, dependence on pH and ionic strength), complex synthesis and storage/operational procedure [28,44,45] exist. Given the importance and specificity of PEX domain in MMPs, enhancing response time and storage/operational stability while lowering cost and maintaining selectivity is highly attractive because reliable and cheap point-of-care testing diagnostics are needed to respond to cancer. In this work, we proposed a new avenue to detect MMP-14 by applying a synthetic chemical inhibitory chemical molecule as the recognition element. Our hypothesis was that a non biocomponent would bind specifically with the PEX domain of MMP-14, overcoming the limitations related to the use of unstable biomolecules and the inaccessible catalytic domain. A simple therapeutic would support low cost, reproducibility, resistance in less favorable microenvironments and stability in prolonged storage [46,47]. 

In this work, we engineered a fast, precise, and specific electrochemical MMP-14 sensor by targeting the PEX like domain of MMP-14 using a therapeutic ligated onto multi-walled carbon nanotubes (CNTs). Among the different inhibitory therapeutic, a synthetic molecule, NSC-405020 (3,4-Dichloro-N-(pentan-2-yl) benzamide) was chosen because of its single action on the PEX domain of MMP-14. NSC 405020 is a specific noncatalytic inhibitor of MMP-14, that directly interacts with PEX domain, affecting protein homodimerization but not the catalytic activity [40,48]. CNTs offer excellent intrinsic properties such as high surface area, chemical stability and high electrical conductivity (10^6^–10^7^ S/m) [49], becoming one of the most attractive nanomaterials in electrochemical sensing [50,51,52]. Furthermore, CNTs act as signal amplifier due to their high specific area that allows multitude of proteins to be gathered [39]. The resultant inhibitor loaded CNT system was physiochemically characterized and tested in PBS (pH 7.40) to verify the ability to detect specifically MMP-14 and not other metalloproteinase. 

## 2. Methodology

### 2.1. Functionalization of CNTs

Multi walled carbon nanotubes (CNTs) were functionalized in 3 steps: (i) oxidation in an acid mixture [53]; (ii) generation of acyl chloride functional groups by suspension in a solution of SOCl_2_ [54]; and (iii) covalent grafting of the small molecule [55] (Figure 1). Briefly, in the first step, 50 mg of CNTs (Sigma-Aldrich, Oakville, ON, Canada) were added to 50 mL of H_2_SO_4_:HNO_3_ (3:1) (Fisher Scientific, Ottawa, ON, Canada) mixture, dispersed for 2 h in an ultrasound bath and then, upheld for 15 h. After, HCl (Fisher Scientific, ON, Canada) was added to the solution. Subsequently, ammonium hydroxide (Fisher Scientific, ON, Canada) was used to neutralize. Finally, oxidized CNTs were filtered using a 0.22 µm nylon membrane (GVS, Fisher Scientific, ON, Canada), washed with ultra-pure water (Milli-Q, Merck, Darmstadt, Germany) until neutral pH and dried overnight in an oven at 40 °C. The second step consisted of the generation of the acyl chloride on CNTs surfaces. The oxidized CNTs were suspended in SOCl_2_ (0.1 g of CNT per 20 mL of SOCl_2_) (Fisher Scientific, ON, Canada) and dispersed for 20 min in an ultrasound bath. The solution was stirred at 70 °C for 36 h. The resulting acylated CNTs were filtered, washed several times with anhydrous tetrahydrofuran (Fisher Scientific, ON, Canada) and dried overnight at 40 °C. 3,4-Dichloro-N-(pentan-2-yl) benzamide, (MMP-14 Inhibitor- NSC405020 (AmBeed, Arlington Heights, IL, USA)) was grafted onto acylated CNTs. The inhibitor molecule was mixed with 1 mL solution of DMF (Fisher Scientific, ON, Canada) and NaH (60%) (Fisher Scientific, ON, Canada) and then stirred for 1 h. The obtained acylated CNTs were then added to the suspension (molecule-to-CNTs weight ratio 15:1). The reaction was kept at 100 °C for 5 days. After, the inhibitor loaded CNTs were filtered, washed several times with ultra-pure water and dried overnight at 40 °C. 

### 2.2. Materials Characterization

The morphology of pristine CNTs, acylated CNTs, and inhibitor loaded CNTs was investigated by Scanning Electron Microscopy (SEM, Inspect F50, FEI Company, Hillsboro, OR, USA). Particle diameter was measured with ImageJ software taking the average of 10 CNTs. Fourier transform infrared spectroscopy (FT-IR–PerkinElmer) was carried out in the wavenumber range of 3500 to 500 cm^−1^ to confirm the grafting of the molecule on CNTs. Cyclic voltammetry (CV) was performed to assure the modification of CNTs from −0.2 V to +0.6 V at scan rate of 10 mv⋅s^−1^. Electrochemical experiments were performed using a potentiostat (VersaSTAT 4, Princeton Applied Research, Oak Ridge, TN, USA) with a three-electrode system cell, where glassy carbon electrode (GCE), acylated CNTs and inhibitor loaded CNTs were used as work electrodes. Platinum wire was used as the counter electrode, and saturated calomel electrode (SCE) as the reference electrode. To prepare the acylated and inhibitor loaded CNTs electrodes, 50% of ethanol (Fisher Scientific, ON, Canada) aqueous solutions containing 1 mg⋅mL^−1^ of acylated CNTs or inhibitor loaded CNTs were prepared and stored. GCE (5 mm, Alfa Aesar, MA, USA) were polished with 0.05 µm Al_2_O_3_ (Fisher Scientific, ON, Canada) suspension to achieve a shiny surface. GCEs were cleaned by sonication in 10% H_2_SO_4_ (Fisher Scientific, ON, Canada), 50% acetone (Fisher Scientific, ON, Canada), and ultra-pure water -each for 10 min successively. The electrodes were dried at room temperature. Finally, 20 µL of the desired solution was dropped onto the cleaned GCE and dried at room temperature.

### 2.3. Detection of MMP-14

The detection of the MMP-14 protein was performed via electrochemical impedance spectroscopy (EIS) within the frequency from 50 kHz to 500 Hz, 12 points per decade, applied potential of 50 mV, and direct potential of +0.20 V. All CVs and EIS spectra were obtained in PBS (pH 7.40) containing 10 mmol.mL^−1^ of K_4_[Fe (CN)_6_]^−^ (Sigma-Aldrich, ON, Canada), 10 mmol⋅mL^−1^ of K_3_[Fe (CN)_6_]^−^ (Sigma-Aldrich, ON, Canada) and 10 mmol.mL^−1^ of NaCl (Fisher Scientific, ON, Canada) [22]. MMP-14 and MMP-1 proteins were unfolded prior experiments. Briefly, 1 µg⋅mL^−1^ of protein was incubated for 1 h at 37 °C with 200 µM EDTA (Fisher Scientific, ON, Canada) and 5 mM 2-Mercaptoethanol (Sigma-Aldrich, ON, Canada) to generate denatured MMPs [56]. Afterwards, solutions with the desired concentration (from 0 to 250 ng⋅mL^−1^) were prepared in PBS (pH 7.4), dropped onto electrodes surfaces (50 µL), and incubated at room temperature for 10 min. Electrodes were then extensively rinsed with ultra-pure water to remove any physically adsorbed MMP-14. EIS was performed in presence and absence of the proteins, applying the acylated and the inhibitor loaded CNTs electrodes. Circuit fit was performed using EC-Lab^®^ software (Version 10.38, Biologic Science Instruments, France) (Appendix A). Resistance to charge (Rct) was extracted from the intercept on the real axis.

Limit of detection (*LOD*) was calculated according to the Formula (1): 
(1)
$$LOD=3\times \left(\frac{SD}{S}\right)$$

where, *SD* is the standard error intercept and the *S* is the slope of the calibration curve (*S*), both extracted from origin software after linear regression. 

### 2.4. Statistics

The data were analyzed using OriginPro (OriginLab Corporation, version 2018G’ MA, USA) and presented as mean ± SD. A one-way ANOVA and Tukey test were performed to evaluate statistical significance. *p*-Value smaller than 0.05 denotes significant difference. 

## 3. Results and Discussion

To overcome the challenging access to the catalytic core of the protein, we were interested to investigate and detect the PEX domain. Here, CNTs were chemically modified with PEX like domain inhibitory molecule to specifically bind one of MMP-14 domains. Pristine carbon materials are usually chemically inert, and require a prior surface treatment for activating and facilitating the immobilization of molecules of interest to offer a durable grafting and so a long term use [57]. CNTs were initially oxidized by a room temperature process in acid mixture of HNO_3_ and H_2_SO_4_, with addition of HCl. Among the available methodologies to produce oxidized carbon structures, the designated for this work has been shown as the best to enable a high percentage of hydroxyl and carboxyl groups on CNTs surface [53]. The generated carboxylates were then further modified to more reactive acyl chloride groups after reacting with SOCl_2_. After acylation reaction, the inhibitory molecule NSC 405020 was covalently bounded to the acylated CNTs surface, via the addition-elimination process of the acyl group, resulting in the inhibitor loaded CNTs (Figure 1A). The covalent bond assures a more robust and stable connection between the organic compound and the CNTs in comparison to noncovalent procedures, generating a more effective electrode interface [29,58,59,60].

SEM images showed mild changes on the surface of CNTs after chemical treatment at high magnification (Figure 1B). Due to van der Waals’ attraction causing significant agglomeration, the pristine CNTs show a longer length compared to the modified CNT [59,60]. As expected, after oxidation and generation of acyl groups the acylated CNTs showed a slight reduction in the length of the nanotubes [52,58]; however, they are denser and less aggregated due to the repulsion between CNTs with chemical groups attached to their surfaces–allowing a better dispersion [58,59]. The inhibitor loaded CNTs changed notably the dispersion with clearly more aggregated CNTs. The attachment of organic compounds on CNTs surface also increased the tube diameter (36.08 ± 3.55 nm compared to pristine 18.05 ± 1.11 nm) [61,62]. 

Chemical grafting was confirmed with FT-IR analysis and cyclic voltammetry (CV). The FTIR spectra of pristine, acylated, and inhibitor loaded CNTs are shown in Figure 2A. As shown in Figure 2A, pristine CNTs did not show strong peaks when compared to the acylated and inhibitor loaded CNTs. Weak intensity could be observed at ~2105, 1500, 1267 and 1007 cm^−1^, typical peaks related to the C=C bond from the hexagonal CNT structure [59]. After oxidation and generation of acyl groups, weak peaks were observed at ~1813 and 1700 cm^−1^, that could be attributed to C=O bond stretching vibration as a result of carbon oxidation [53,54,59,63]; The peak at ~1388 and 1000 cm^−1^ is associated with C–O vibrations from acyl groups [59,64].

After grafting the inhibitor on CNTs, additional peaks were observed at ~3012, 1513 and 1412 cm^−1^, assigned to C–C and C=C vibration of the benzene ring [65,66,67]. The strong peak at ~675 cm^−1^ is characteristic of C–Cl bond present in the chlorobenzene [65,68]. The peak at ~1704 cm^−1^ is associated with N–H stretching from the molecule structure [69]. Note that because of the grafting process leading to change in electronegativity of the neighboring atom, some peaks have shifted from the expected wavenumber, such as C–Cl (~730 cm^−1^) and N–H (~1610 cm^−1^). Cyclic voltammetry was carried out with ferrocyanide/ferricyanide redox couple (Fe (CN)_6_^3−/4−^) to confirm the presence of inhibitor on CNTs (Figure 2B). An increase in peak current density (*I*_pa_ = ~10.30 µA), and a decrease of peak width (Δ*E*_p_ = ~134 mV) was observed in the Fe (CN)_6_^3−/4−^ voltammograms after deposition of CNTs on GCE because of the higher active surface area. Acylated CNT presented a significant improvement in the peak current density (*I*_pa_ = ~40.45 µA), and a decrease of peak width (Δ*E*_p_ = ~105 mV). It is known that the carbon oxidation and the generation of acyl groups improve the high electron transfer rate by introducing negative charges onto CNTs surface and increasing the number of active sites on electrode surface [52,60]. As expected, CNTs modified with the inhibitory molecule caused a significant drop of peak current density (*I*_pa_ = ~12.30 µA), attributed to a loss of conductivity. These results were consistent with SEM and FTIR, confirming the successful grafting of molecule onto CNTs surface. 

The efficacy of the inhibitor loaded CNTs towards the detection of MMP-14 was investigated using electrochemical impedance spectroscopy (EIS). Figure 3A shows the impedance spectra of the inhibitor loaded CNT electrode for the blank and MMP-14 in concentrations of 10, 50 and 100 ng⋅mL^−1^, and its respective limit of detection (LOD) (Figure 3B). Note: logarithmic values were applied in Appendix A. 

At high MMP-14 concentrations, the real and imaginary part of the impedance increased drastically; the electrochemical signal is reproducible and stable during each measurement for the same concentration of MMP-14. A linear relationship between MMP-14 concentration and the charge transfer resistance (R_CT_) values was observed. The inhibitor loaded CNT electrode achieve a high sensitivity of 2.83 µA⋅log [MMP-14]^−1^ and a linear correlation of 0.99. The designed inhibitor loaded CNT system presented LOD of 7.5 ng⋅mL^−1^, and range of detection from 10 to 100 ng⋅mL^−1^ (Appendix A). Most common electrochemical systems applied to detect MMPs up to now have been using carbon and gold as substrate [39,57] combined with biomolecules, such as aptamers, antibodies, enzymes, or peptides as the recognition element [28]. Carbon-Gold-Pb-Peptide sensor was applied for the detection of MMP-2 [70]. The sensor exhibited linear detection from 1 pg⋅mL^−1^ to 1 μg⋅mL^−1^, high sensitivity of 28.4 μA⋅log [MMP-2]^−1^ and low LOD of 0.40 pg⋅mL^−1^ but was fabricated with peptides through several steps in a complex synthesis using polyaniline gel as substrate and CS-AuNPs-Pb^2+^ as impedance enhancer. In a recent study, silver nano particles were combined to peptides and receptor to allow the detection of MMP-2 by anodic stripping voltammetry [71]. The method offered a significant improvement when compared to the traditional ELISA assay, presenting range of detection from 0.5 pg⋅mL^−1^ to 50 ng⋅mL^−1^ and LOD of 0.12 pg⋅mL^−1^. Besides innovative approach, the presence of bioelements is still a challenge to assure low cost and long stability. Vertically aligned single-wall CNTs were used as substrate to attach antibody and enzyme for the detection of MMP-3, presenting linear detection from 0.4 to 40 ng⋅mL^−1^, sensitivity of 77.6 nA⋅log [MMP-3]^−1^ and LOD of 4 pg⋅mL^−1^ [72]. Despite great sensitivity, the labelling approach applied consumes time, involves complex sample handling and is expensive [44]. Peptide decorated gold-CNT electrode recognizes the MMP-7 protein, presenting linear detection from 0.01 to 1000 ng⋅mL^−1^ and LOD of 6 pg⋅mL^−1^ [73]. A screen-printed electrode was designed by grafting antibodies into 2D structures, graphene oxide and MoS_2_, aiming to amplify the analytical signal for the detection of MMP-7 [74]. Indeed, the 2D nanostructured immunosensor showed improvements and presented range of detection from 10 pg.mL^−1^ to 75 ng⋅mL^−1^ and LOD of 7 pg⋅mL^−1^. 2D graphene structures were also employed to detect MMP-1 protein in an immunosensor based on a gold nanoparticle, polyethyleneimine and reduced graphene oxide [75]. The system showed great performance in different body fluids with range of detection from 1 ng⋅mL^−1^ to 50 ng⋅ml^−1^ and LOD of 0.2 ng⋅ml^−1^. However, the graft of antibodies into 2D graphene structures in both works did not overcome the limitations of immunosensors, including high cost and issues with stability. Only one approach has explored the detection of MMP-14 through the electrochemical interaction with hemopexin domain. In both works, inhibitory thiolated PEX-14 binding peptides (named ISC) were attached to gold electrodes exhibiting a fairly good specificity towards MMP-14, presenting linear detection from 0.4 pg⋅mL^−1^ to 0.05 ng⋅mL^−1^ with LOD of 7 pg⋅mL^−1^ after 30 min; despite the ability of detect very low concentration of MMP-14, the challenges discussed previously, such as instability and complexity of bio elements, remains present. An important note to compare the present work to the ISC peptide is the limited concentration range of detection showed by the peptide system. The designed inhibitor loaded CNT system, presented in this work, revealed as an alternative to the conventional recognized bio elements, with ability to detect MMPs in a clinically relevant concentration range, presenting low cost, reproducibility, and stability in a real-world microenvironment (the loaded CNT system showed stability within 6 months of the study when stored at room temperature). The inhibitor loaded CNTs could be applicable for a variety of cancers as shown in Table 1. The LOD of 7.5 ng⋅mL^−1^ includes the possibility of quantification of MMP-14 levels not only in the cancerous cells but also in different body samples, such as serum and tissues. Note: Cancerous tissue weight varies from ~10 mg to ~48 g depending on the anatomic site [76]. The present study showed the ability of the loaded CNTs in the detection and quantification of MMP-14 in a standard PBS solution, without interference of other biomolecules. Further analysis in biologic samples as such as tissue, serum and cells should be evaluated.

Existing MMP inhibitors target the hidden active center in the catalytic domain and, as a result, impacts activity of many MMPs instead of MMP-14 alone [40]. The interest in detecting the PEX domain is not only because PEX domain is easily accessible but also for selectivity purpose. Indeed, the inhibitory molecule selected for this study has demonstrated specificity for MMP-14 PEX domain, not affecting the catalytic or other MMP PEX domains [40,48]. The small inhibitor interacts at the surface of proteins in a specific region named binding pocket, where small-molecules bind easily with little energetic cost [77]. The conformation changes associated with inhibitor-protein binding is specific to every single protein. Few proteins present null (pre-formed pocket) to minor structural changes to revel the binding pocket, easily allowing the interface with small inhibitors; while others exhibit extensive structural rearrangement in order to establish interactions with the inhibitors through the binding pocket [77]. In this work, we supposed that the exclusivity of inhibitor/protein interaction might be due to that the small molecule NSC405020 is in the particular binding pocket of MMP-14, shaped by Met-328, Arg-330, Asp-376, Met-422, and Ser-470 [40] (Figure 4A). By taking advantage of the inhibitory molecule/protein binding interaction in the PEX domain of MMP-14, a layer of protein was accumulated around CNTs, hampering the electron transfer between the electrode surface and the analyte (Figure 4B). As a result, the resistance of the system increased linearly as a function of the protein concentration (Figure 3B). 

The specificity of the system was further confirmed in presence of MMP-1, another PEX domain based MMP which presents abnormal expression in different types of cancer [78,79]. The EIS response obtained in the presence of 50 ng⋅mL^−1^ of protein is shown in Figure 5. Clearly, no noticeable change was observed, demonstrating the specificity of the selected molecule as the recognition element to interact to the binding pocket of MMP-14. Note that the performance of acylated CNTs without the molecule attached was also verified and no obvious resistance was observed (Appendix A). The inhibitor loaded CNTs system demonstrated selectivity, microenvironmental stability (room temperature and PBS) and low cost when compared to the conventional biosensors. 

## 4. Conclusions

In this work, we focused on the detection of MMP-14 protein by targeting the PEX domain instead of the conventional catalytic domain. MMP-14 is overexpressed in a variety of diseases, including cancer. By taking advantage of the specificity and binding properties of an inhibitory small organic molecule, NSC 402050, an inhibitor loaded CNT system was designed. Pristine CNTs were modified by 3 steps: oxidation, addition of acyl groups and, molecule attachment. The successful grafting was well established by FT-IR, CV, and SEM analysis. The inhibitor loaded CNTs system performance was verified in MMP-14 (PBS, pH 7.4) and demonstrated LOD of 7.5 ng⋅mL^−1^. The specificity was also demonstrated against MMP-1, a MMP type protein that contains PEX domain and is present in abnormal levels in different types of cancer. From our knowledge, this is the first time that an inhibitory small molecule was applied as the recognition element, overcoming the challenges offered by the regular bioelements, such as enzymes, peptides, aptamers, and antibodies. The inhibitor loaded CNT is a desirable candidate to become an alternative method to the complex and expensive diagnostic tools currently available for the detection of MMP-14. Further studies should be performed to conclude the interference with different biomolecules in a real clinical environment (body fluids, tissues, cells, etc.). 

## Figures and Tables

**Figure 1 biosensors-12-00884-f001:**
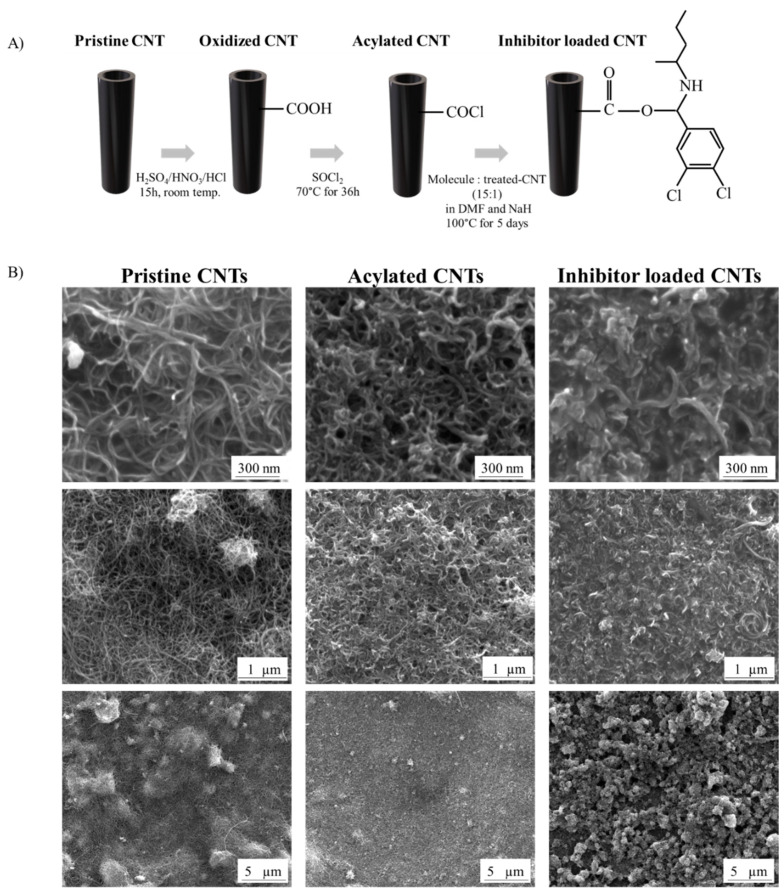
(**A**) Schematic representation of NSC405020 molecule grafted onto CNTs surface followed by (**B**) SEM images for pristine, acylated, and inhibitor loaded CNTs (low and high magnification).

**Figure 2 biosensors-12-00884-f002:**
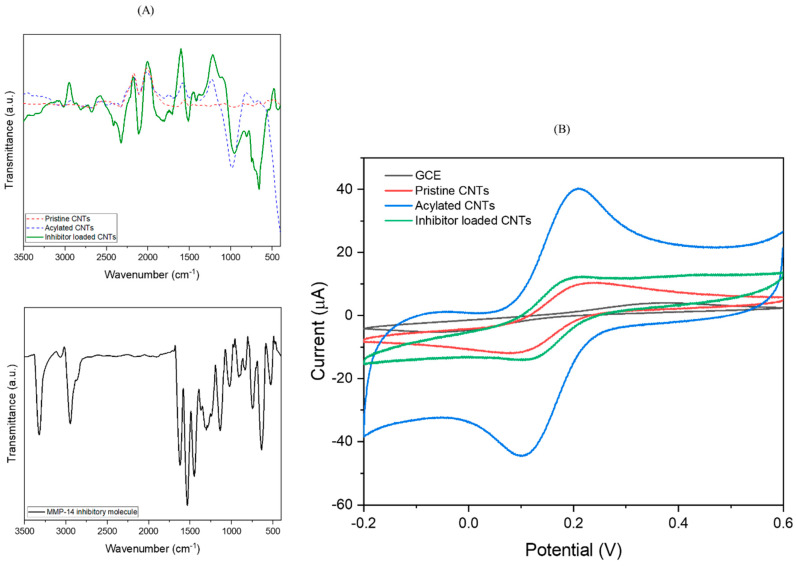
(**A**) FTIR spectrum (top) of pristine, acylated, and inhibitor loaded CNTs, respectively. The spectrum (down) of NSC 405020 molecule is also demonstrated. (**B**) CVs for GCE, pristine, acylated, and inhibitor loaded CNTs in PBS (pH 7.40) containing 10 mmol⋅mL^−1^ of K_4_[Fe (CN)_6_]^−^, 10 mmol mL^−1^ of K_3_[Fe (CN)_6_]^−^ and 10 mmol.mL^−1^ of NaCl, scan rate 10 mV.s^−1^.

**Figure 3 biosensors-12-00884-f003:**
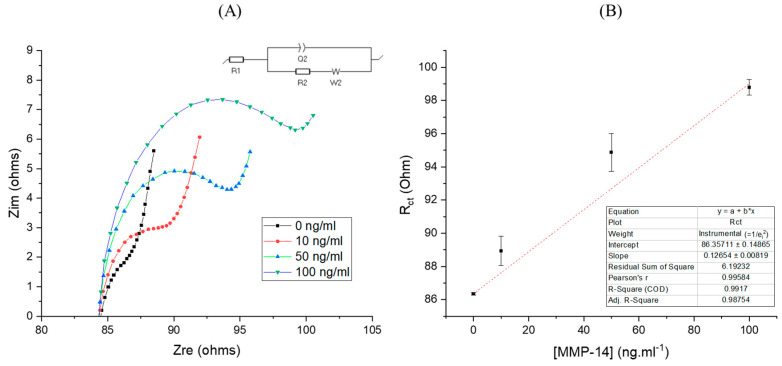
(**A**) Nyquist plots of inhibitor loaded CNTs before and after interaction with different concentrations of MPP-14 for 10 minutes in PBS (pH 7.40) containing 10 mmol.mL^−1^ of K_3_[Fe (CN)_6_]^−^ and 10 mmol.mL^−1^ of NaCl. (**B**) Linear fit of EIS response for different concentrations of MMP-14, presenting LOD of 7.5 ng⋅mL^−1^. Significant difference between measurements was obtained at *p*-value < 0.05 (n = 2). Protein concentration varying from 10 ng.mL^−1^ to 100 ng.mL^−1^, applied potential of +0.20 V, from 50 kHz to 500 Hz, amplitude 50 mV.

**Figure 4 biosensors-12-00884-f004:**
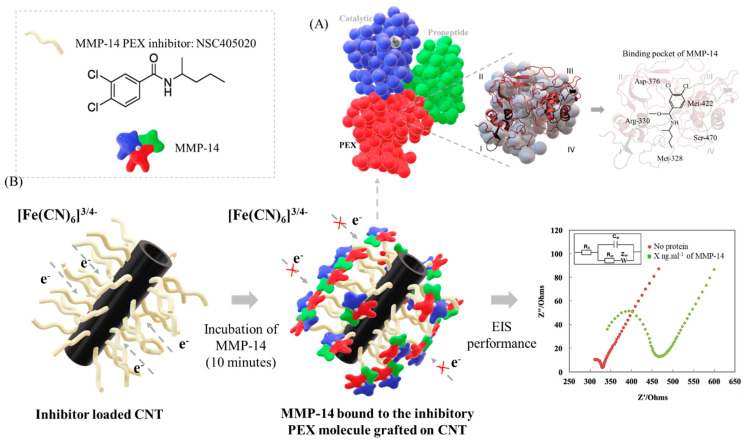
(**A**) The inhibitor/protein interaction between small molecule NSC 405020 and the binding pocket of MMP-14 and (**B**) Schematic of the MMP-14 detection through the binding mechanism of protein and inhibitory small molecule as the recognition element.

**Figure 5 biosensors-12-00884-f005:**
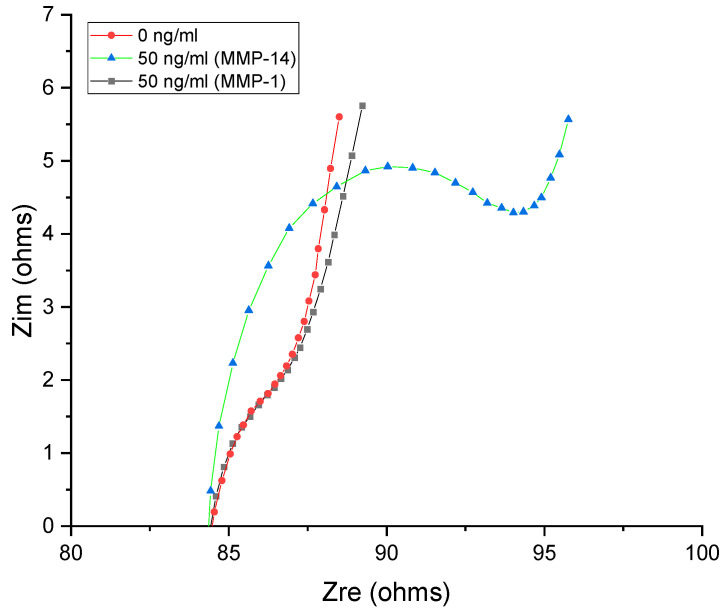
EIS response for the inhibitor loaded CNT after individual interaction with 50 ng⋅mL^−1^ of MMP-1 and MMP-14 in PBS (pH 7.40) containing 10 mmol⋅mL^−1^ of K_4_[Fe (CN)_6_]^−^ 10 mmol⋅mL^−1^ of K_3_[Fe (CN)_6_]^−^ and 10 mmol⋅mL^−1^ of NaCl. The inhibitor loaded CNT showed specificity to MMP-14 (applied potential of +0.20 V, from 50 kHz to 500 Hz, amplitude 50 mV).

**Table 1 biosensors-12-00884-t001:** Abnormal expression of MMP-14 for different types of cancer and body samples.

Type of Cancer	Sample	MMP-14 Level (Healthy)	MMP-14 Level (Non-Healthy)	Ref.
**Breast**	Serum Cell	8.55 ± 1.66 ng⋅mL^−1^ -	16.91 ± 5.87 ng⋅mL^−1^ ~26.7 ng⋅mg^−1^ *	[32,33]
**Head & Neck**	Tissue Cell	0.80 ± 1.1 ng⋅mL^−1^ -	5.00 ± 4.3 ng⋅mg^−1^ ~14.5 ng⋅mg^−1^ *	[33]
**Fibrosarcoma**	Tissue Cell	<0.01 ng⋅mg^−1^ * -	~1.12 ng⋅mg^−1^ * ~36.1 ng⋅mg^−1^ *	[33,34]
**Prostate**	Tissue	<0.01 ng⋅mg^−1^ *	0.6 ± 0.05 ng⋅mg^−1^	[34]
**Gastric**	Tissue	~1.34 ng⋅mg^−1^ *	~3.57 ng⋅mg^−1^ *	[35]
**Bladder**	Tissue	7.45 ± 1.18 ng⋅mg^−1^	81.78 ± 9.87 ng⋅mg^−1^	[36]
**Ovarian**	Serum	~6 ng⋅mL^−1^ *	~12 ng⋅mL^−1^ *	[37]
**Brain**	Tissue	~1 ng⋅mg^−1^ *	~10 ng⋅mg^−1^ *	[38]

“-”: data not provided; “*”: Study did not provide the SD value.

## Data Availability

Not applicable.

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
