# Peer review of "Detecting the PEX Like Domain of Matrix Metalloproteinase-14 (MMP-14) with Therapeutic Conjugated CNTs"

_biosensors, 2022, doi:10.3390/bios12100884_

Round 1

Reviewer 1 Report

This manuscript should be improved.

Specific comments are as follows:

1.     Title. It can be read out from the title that the goal of this study if the detection of MMP-14.

2.     Abstract. From “: Matrix metalloproteinases (MMPs) are” to “making the electrochemical response very challenging”. More than half of the Abstract section is talking about the problem statement. It is not accepted.

3.     Introduction. This section should be revised with the introduction of recent studies with the citation of recent papers.

4.     So limited methods for materials characterization are used. The detection of MMP-14 should be stated as section 2.3. In addition, the way for the calculation of Rct from EIS data, and the way for the calculation of LOD in this study should be introduced.

5.     The performance of this study should be compared with previous reports.

6.     The application for practical samples and the effects of environmental parameters should be evaluated.

7.     Only MMP-1 was used for the selectivity evaluation.

8.     The performance of storage or reusability were not tested.

Author Response

Reviewer 1  

This manuscript should be improved.

Thank you for your comments and suggestions

  1. Title. It can be read out from the title that the goal of this study if the detection of MMP-14.

Indeed, our aim is to detect MMP-14 through its PEX domain.

  1. Abstract. From “: Matrix metalloproteinases (MMPs) are” to “making the electrochemical very challenging”. More than half of the Abstract section is talking about the problem statement. It is not accepted.

Abstract was rewritten.

  1. Introduction. This section should be revised with the introduction of recent studies with the citation of recent papers.

We reviewed the introduction, and all modifications are tracked in the manuscript. Some citations were updated.  

  1. So limited methods for materials characterization are used. The detection of MMP-14 should be stated as section 2.3. In addition, the way for the calculation of Rct from EIS data, and the way for the calculation of LOD in this study should be introduced.

Materials characterization includes morphology and topography with SEM, chemical attachment of the inhibitor molecule and modification of CNTs with FTIR and cyclic voltammetry. All these characterizations support the modification of the CNT and the expected results confirmed by EIS analysis.

We improved the methodology and added details regarding the Rct and LOD calculation for the detection of MMP-14.   

  1. The performance of this study should be compared with previous reports.

There is currently one approach to detect hemopexin domain of MMP-14 through peptides. The work was cited in the introduction (ref 22 and 43), but we also incorporated into the discussion and explained the limitation of the detection range in a clinical scenario (page 14).  

  1. The application for practical samples and the effects of environmental parameters should be evaluated.

We agree with the reviewer that further investigations will be required for clinical implementation.  This paper is about a new design and a proof of principle that a commercial drug can act as the recognition element in the detection of MMP14. Experiment in PBS is standard procedure and use traditionally in practices after protein extraction from cells or tissues. This was justified in the discussion and conclusion the information. Using in vivo model or human tissue with ethic approvals combined with a screen printing electrode or another electrochemical system more adapted to tissue should be the next step but we believe, is out of the scope for this manuscript.  

  1. Only MMP-1 was used for the selectivity evaluation.

Given that MMP-1 is overexpressed in cancerous tissues and cells and most importantly presents the hemopexin domain in its structure, the electrode specificity was critical to assess in presence of MMP-1. This study was to show that the drug only interacted with the binding pocket of MMP-14 and not within the MMP-1 environment. We clarified this in the discussion.     

  1. The performance of storage or reusability were not tested.

The storage capability was tested during 6 months at room temperature and demonstrated efficacy within this time. Longer storage (above 6 months) was not tested. We added this information in the discussion. Detection is based on the physical interaction between the drug and amino acids in the binding pockets of MMP14.  Regenerability of the electrode could be envisaged with detergent or temperature, however, as we used low cost of materials (drug and carbon) to fabricate the electrode, we envisioned the device to be disposable. This was clarified in the discussion p15. 

Reviewer 2 Report

The manuscript of Vieira et al. is focused on the development of biosensor for the detection of MMP-14 protein. It is based on the preparation of the biorecognition element composed of an artificial inhibitor combined with carbon nanotubes, which is recognized by the determined protein via interaction with the non-active domain (called PEX domain) instead of the catalytic one. The designed biosensor is simple, affordable and can be used routinely

The manuscript is divided into the usual parts, Introduction, Methodology and Results which are appropriately discussed.

Before it will be acceptable for publication in Biosensors it needs major revision.

1.       The title used does not accurately describe the content of the article. The work consists in the development of a biosensor for protein detection and quantification, not only for verifying the interaction of modified CNTs with the analysed protein, as stated. Therefore, I recommend to modify the title to fulfil the exact focus of article.

2.       Abstract should be modified to be much more focused on the main points and achieved results. It contains duplication in the statement that is given in the introduction.

3.       The explanation of abbreviation ELISA used in abstact is missing. For abbreviation CNTs it is the same.

4.       On Figure 1A – the complete structure (functional group included in reaction) of inhibitor should also be mentined for better clarity.

5.       On Figure 2 – descriptions of curves are not readable – compare to voltammogram. Also, legend, the position of FTIR spectrum (top, down) should be written.

6.       On Figure 3B - in my opinion, linear interpolation is not adequate to the stated values. I recommend to use logarithmic values on x axis. Then, linear regression will be adequate. Also, the text in table is not well readable.

7.       On page 7, line 262 authors write about the instability of biosensors prepared with use of antibodies, aptamers or other. Especially, in citation 67 – the stability is really good, not bad, as are mentioned here (line 255, same page). Moreover, the authors discuss the stability, but in this article the stability of developed biosensor was not tested. I recommend to complete these data.

8.       In whole article in units “ml” should be replaced by “mL”.

9. In Figure 4 - notes - text - are not well readable. 

Author Response

Reviewer 2

The manuscript of Vieira et al. is focused on the development of biosensor for the detection of MMP-14 protein. It is based on the preparation of the biorecognition element composed of an artificial inhibitor combined with carbon nanotubes, which is recognized by the determined protein via interaction with the non-active domain (called PEX domain) instead of the catalytic one. The designed biosensor is simple, affordable and can be used routinely

The manuscript is divided into the usual parts, Introduction, Methodology and Results which are appropriately discussed. 

Thank you for your comments and suggestions

  1. The title used does not accurately describe the content of the article. The work consists in the development of a biosensor for protein detection and quantification, not only for verifying the interaction of modified CNTs with the analysed protein, as stated. Therefore, I recommend to modify the title to fulfil the exact focus of article.

We modified the title.

  1. Abstract should be modified to be much more focused on the main points and achieved results. It contains duplication in the statement that is given in the introduction. 

Abstract was modified.

  1. The explanation of abbreviation ELISA used in abstact is missing. For abbreviation CNTs it is the same. 

Abstract was modified and abbreviations added in the manuscript.

  1. On Figure 1A – the complete structure (functional group included in reaction) of inhibitor should also be mentined for better clarity.

The inhibitor is in Figure 1A with functional group. We clarified in the caption.

  1. On Figure 2 – descriptions of curves are not readable – compare to voltammogram. Also, legend, the position of FTIR spectrum (top, down) should be written.

Corrected and added.

  1. On Figure 3B - in my opinion, linear interpolation is not adequate to the stated values. I recommend to use logarithmic values on x axis. Then, linear regression will be adequate. Also, the text in table is not well readable.

We used linear regression to align our data with previous works especially, the ones using similar approach with PEX domain electrochemical detection (reference 22 and 43), this allow us to compare similar responses, but if requested by reviewer we will add this data in supplementary information.  Text size was increased in Fig. 3B.

  1. On page 7, line 262 authors write about the instability of biosensors prepared with use of antibodies, aptamers or other. Especially, in citation 67 – the stability is really good, not bad, as are mentioned here (line 255, same page). Moreover, the authors discuss the stability, but in this article the stability of developed biosensor was not tested. I recommend to complete these data.

Antibodies are presumably the most important and prominent class of biorecognition elements, however, in terms of stability, they are limited to physiological conditions and prone to irreversible denaturation. Aptamers have enhanced stability compared to antibodies due to their temperature tolerance and other experimental conditions, but both have limited storage conditions, usually limited to several days, under refrigeration and wet conditions. We emphasize these points in introduction  Reference 67 (now 70) exhibits indeed a good stability for biomolecules but not as superior as the use of a relatively inert organic molecule that can be stored for 6 months on the shelf at room temperature. We corrected the citation to “Carbon-Gold-Pb-Peptide sensor was applied for the detection of MMP-2 [70]. The sensor exhibited linear detection from 1 pg.mL-1 to 1 μg.mL-1, high sensitivity of 28.4 μA.log [MMP-2]-1 and low LOD of 0.40 pg.mL-1 but was fabricated with peptides through several steps in a complex synthesis using polyaniline gel as substrate and CS-AuNPs-Pb2+ as impedance enhancer” and discussed the stability of our sensor.

  1. In whole article in units “ml” should be replaced by “mL”.

Corrected.

  1. In Figure 4 - notes - text - are not well readable. 

Text sizes were increased in Figure 4.

Reviewer 3 Report

Ligating the PEX like domain of Matrix Metalloproteinase-14 2 (MMP-14) with therapeutic conjugated CNTs 

Biosensors

Thank you for asking me to review the above-titled manuscript. The topic is interesting. However, there are problems that need to be addressed.

Abstract: 1) Add a research question. 2) Give some figures in regard to sensitivity, specificity, Positive PV, and negative PV of your approach versus the currently used method. 3) not clear in the abstract if human tissues/cells or animals were used. 4) Show the full words of CNTs in the abstract.

Introduction: 1) State a research question. 2) Justify the scientific basis for your method and your approach. What are the limits of the currently used method/s?

Methods: 1) The authors may need a subtitle after number 2.2 and before statistics (to become 2.4) to discuss the actual experimental done and what exactly was performed and the comparison between the new approach and the currently available method. 2) Methods need references.

Results and Discussion: 1) It will be of interest to show more the sensitivity, specificity, and positive and negative PVs, of your approach compared to the currently used method, show figures. 2) Also discuss the advantages and limitations of your method. 3) These changes should be shown in the results, discussion and conclusion. 

Others- 1) Add a list at the end showing the full words of all abbreviations. 2) Follow the journal guidelines as you review your manuscript. 3) Some editing is needed for the manuscript to make things more straightforward. 

Author Response

Reviewer 3

Thank you for asking me to review the above-titled manuscript. The topic is interesting. However, there are problems that need to be addressed.

Thank you for your comments and suggestions

Abstract: 1) Add a research question. 2) Give some figures in regard to sensitivity, specificity, Positive PV, and negative PV of your approach versus the currently used method. 3) not clear in the abstract if human tissues/cells or animals were used. 4) Show the full words of CNTs in the abstract. 

Abstract was updated and corrected.

Introduction: 1) State a research question. 2) Justify the scientific basis for your method and your approach. What are the limits of the currently used method/s? 

Introduction was reviewed and comments were all addressed and added. The modifications are all tracked on the manuscript.

Methods: 1) The authors may need a subtitle after number 2.2 and before statistics (to become 2.4) to discuss the actual experimental done and what exactly was performed and the comparison between the new approach and the currently available method. 2) Methods need references.

Methods section was divided in subtitles: 2.1 Functionalization of CNTs; 2.2 Materials characterization; 2.3 Detection of MMP-14; and 2.4            Statistics. References were added in methods.

Results and Discussion: 1) It will be of interest to show more the sensitivity, specificity, and positive and negative PVs, of your approach compared to the currently used method, show figures. 2) Also discuss the advantages and limitations of your method. 3) These changes should be shown in the results, discussion and conclusion. 

The performance of the study was compared with similar techniques for the detection of MMPs in general. There is only one approach to detect hemopexin domain of MMP-14 through peptides. The work was cited in the introduction (ref 22 and 40), but we also added to the discussion to make it easier to the audience, and we extended the discussion further, including the limitation of the detection range in a clinical scenario (page 14).

The main limitation of the study would be the interference of the biomolecules, adding the purification step before the analysis. We added the need of further study to the discussion and conclusion. 

Others- 1) Add a list at the end showing the full words of all abbreviations. 2) Follow the journal guidelines as you review your manuscript. 3) Some editing is needed for the manuscript to make things more straightforward

List added. The manuscript guidelines were reviewed and adapted as requested.

MMPs

Matrix metalloproteinases

PEX

Hemopexin

CNTs

Carbon nanotubes

SEM

Scanning Electron Microscopy

FTIR

Fourier transform infrared spectroscopy

CV

Cyclic Voltammetry

EIS

Electrochemical Impedance Spectroscopy

PBS

Phosphate buffered saline

ELISA

Enzyme-linked immunosorbent assay

GCE

Glassy Carbon Electrode

SCE

Saturated Calomel Electrode

LOD

Limit of Detection

SD

Standard error intercept

S

Slope

SD

Standard deviation

Ipa

Peak current density

ΔEp

Peak width

RCT

Charge transfer resistance

ISC

Thiolated PEX-14 binding peptides

Round 2

Reviewer 1 Report

Performance of matrix metalloproteinases (MMPs) detection should be compared with previous studies. In addition, most of the papers cited in this study are published 5 years ago, and should be updated.

Author Response

Thank you for your comments. We updated the literature review in the discussion p15.  The current electrochemical methods to detect MMPs [reference 70 (2019), 71 (2021), 72 (2010), 73 (2020), 74 (2021), 75 (2021), 22 (2019) and 43 (2019)] use bioelements to recognize MMPs, and all these works have been now referenced in the manuscript. 

Reviewer 2 Report

Thans to authors for the manuscript revision, which were beneficial. 

Only one comment to point 6. 

  1. On Figure 3B - in my opinion, linear interpolation is not adequate to the stated values. I recommend to use logarithmic values on x axis. Then, linear regression will be adequate. Also, the text in table is not well readable.

We used linear regression to align our data with previous works especially, the ones using similar approach with PEX domain electrochemical detection (reference 22 and 43), this allow us to compare similar responses, but if requested by reviewer we will add this data in supplementary information.  Text size was increased in Fig. 3B.

I highly recommend to add suggested data at least to supplementary information. Linear regression is inappropriate in this case, although the authors mention the comparability as the reason.

Author Response

Thank you for your comments.

On Figure 3B - I highly recommend to add suggested data at least to supplementary information. Linear regression is inappropriate in this case, although the authors mention the comparability as the reason.

As recommended we added logarithmic values on x axis on the linear regression in supplementary information.